# A Model of Pixel and Superpixel Clustering for Object Detection

**DOI:** 10.3390/jimaging8100274

**Published:** 2022-10-06

**Authors:** Vadim A. Nenashev, Igor G. Khanykov, Mikhail V. Kharinov

**Affiliations:** 1Laboratory of Intelligent Technologies and Modelling of Complex Systems, Institute of Computing Systems and Programming, Saint Petersburg State University of Aerospace Instrumentation, 67 B. Morskaia St., 190000 Saint Petersburg, Russia; 2Laboratory of Big Data Technologies for Sociocyberphysical Systems, St. Petersburg Federal Research Center of the Russian Academy of Sciences, 14 Line V.O. 39, 199178 Saint Petersburg, Russia or

**Keywords:** color image, object detection, pixel clustering, piecewise constant approximations, hierarchical sequence, total squared error

## Abstract

The paper presents a model of structured objects in a grayscale or color image, described by means of *optimal* piecewise constant image approximations, which are characterized by the minimum possible approximation errors for a given number of pixel clusters, where the *approximation error* means the total squared error. An ambiguous image is described as a non-hierarchical structure but is represented as an ordered superposition of object hierarchies, each containing at least one optimal approximation in *g*_0_ = 1, 2,..., etc., colors. For the selected hierarchy of pixel clusters, the objects-of-interest are detected as the pixel clusters of optimal approximations, or as their parts, or unions. The paper develops the known idea in cluster analysis of the joint application of Ward’s and K-means methods. At the same time, it is proposed to modernize each of these methods and supplement them with a third method of splitting/merging pixel clusters. This is useful for cluster analysis of big data described by a convex dependence of the optimal approximation error on the cluster number and also for adjustable object detection in digital image processing, using the optimal hierarchical pixel clustering, which is treated as an alternative to the modern informally defined “semantic” segmentation.

## 1. Introduction

This paper presents interdisciplinary research in cluster analysis of big data and image processing. A mathematical model for detecting objects in an image by Ward’s and two other methods of adaptive hierarchical pixel clustering is proposed. The approach is to approximate the image in different numbers of colors with the minimum approximation error or standard deviation of the approximations from the image. The minimization problem is NP-hard [1]. Therefore, it cannot be solved in practice using a general cluster analysis. In modern image processing, real optimization of pixel clustering is achievable only if there is an adequate specific image model, which so far can only be dreamed of. We develop a solution to the problem of real-life minimization of the approximation error E and simultaneously construct an adequate image model. In this way, we take into account the specificity of pixel sets, which distinguishes them from arbitrary data. While constructing a model for image processing of any content, we do not assume prior learning procedures for detecting objects-of-interest but only allow setting up a software system using predefined parameters.

The main disadvantage of learning systems for image recognition is the training procedure asperities, comparable to the complexity of direct programming the recognition of specific objects. An alternative to learning systems is customizable recognition systems, which have been massively created and continue to be created to solve an expanding range of practical problems. Reducing the complexity of creating and operating customizable recognition systems is achieved mainly due to the active use of a priori information about objects and the problem being solved [2]. At the same time, taking into account a priori information limits the use of software products in changing application conditions, as well as in various tasks. The unification of specialized solutions, as a rule, is hindered by the insufficient consideration of the ambiguity of the image, on which the target objects-of-interest can be visually observed objects of various scales. For this reason, in the field of image recognition, even the simplest tasks such as text recognition remain relevant [3], and software development turns out to be too expensive, which was noticed by American officials long ago [4].

To solve the technical and organizational problems of computer image recognition, it is important to create a transparent model of objects in an image of any content, taking into account the ambiguity of the image and providing for setting up the recognition system for the desired objects.

The aim of the paper is to describe a formal model for detecting objects in an image by modernized versions of classical cluster analysis methods [5,6] (Otsu’s methods [7,8,9,10], Ward’s [1,11,12,13], K-means [1,14,15,16,17,18,19] and combined splitting/merging clustering methods [11,20,21,22]), which reduce the image approximation error of approaching an image by its piecewise constant approximations. In this case, the requirement of minimizing the approximation error is to be fulfilled, which, as a rule, conflicts with the heuristic consideration of a priori information about objects. The requirement of real-life minimization of the approximation error helps to avoid the typical engineering approach to image recognition of previously known content and contributes to the unification of solutions for images of any content.

The main contributions of this work include:We propose a mathematical model for detecting objects in an image, assuming that the sequence of approximation errors of optimal approximations in a successively increasing number of colors is convex.We formalize the concepts of images, objects, and superpixels, distinguishing them from each other by structure to unify object detection without referring to specific examples of images and objects.To verify the object detection model, the criterion of minimum standard deviation σ or approximation error $E\sim \sigma^{2}$ is used instead of using “ground truth” samples that do not take into account image ambiguity.We offer three modernized versions of the classical methods for real-life minimizing approximation errors and customizable object detection by hierarchical image approximations.

The rest of this study is arranged as follows.

Section 2 illustrates the advantage of pixel clustering compared to image segmentation using a standard image as an example.

Section 3, Section 4 and Section 5 describe the image model with the already constructed sequence of optimal approximations. Section 3 gives a transparent definition of superpixels that is independent of a particular algorithm and illustrates this definition with an example of a real-life image. In Section 4, the fundamental features of the proposed superpixel concept in comparison with known solutions are discussed. Section 5 describes a model for detecting the hierarchy of objects and the objects themselves in a digital image, which is based on optimal pixel clustering.

Section 6, Section 7, Section 8 and Section 9 explain how to obtain a sequence of optimal image approximations. Section 6, Section 7 and Section 8 deal with three classical methods of cluster analysis, namely: Ward clustering, split/merge methods, and K-means. Their main shortcomings are indicated and modernized versions are proposed, which calculate optimal image approximations and generate the superpixels. Section 9 explains how the modernized methods for minimizing approximation errors work together in the frameworks of the model for object detection.

Section 10 describes the computer implementation of the model for object detection. Finally, the results and perspectives of this study are summarized in Section 11.

## 2. Pixel Clustering vs. Image Segmentation

Image segmentation, as a result of dividing an image into a relatively small number of connected segments, is rightly treated in the vast majority of papers on object detection as an attribute of the initial image processing. However, this does not at all imply an understanding of segmentation as a process of merging adjacent segments during bottom-up image processing. Obviously, supporting pixel connectivity within the clusters can interfere with minimizing the approximation error. In our experience, this is indeed the case, as illustrated in Figure 1.

Figure 1 on the left shows the optimal approximation of the standard “Lena” image in two levels of intensity, calculated by Otsu’s method [8]. On the right is the approximation obtained by error $E\sim \sigma^{2}$ minimizing using iterative merging adjacent segments. The image approximation on the right is significantly worse than the central image approximation with the standard deviation close to the minimum. The central approximation is obtained if, in the optimal image approximation (on the left), one-color segments are interconnected by thin lines one pixel thick. It turns out that segmenting an image with a real-life minimum approximation error is, in principle, more difficult than solving the same problem for pixel clustering. At one time, this motivated us to move from segmentation to the study of more efficient pixel clustering.

In the theory of clustering, we start from the results of [1] and subsequent works, which are listed in detail in [23]. In these works, the features of Ward’s clustering are studied, and the problem of joint application of Ward’s and the K-means methods is posed. To minimize the approximation error E by the K-means method, it is proposed to pre-calculate the approximation hierarchy using Ward’s method and store it in terms of traditional trees (dendrograms). In this way, Ward’s method provides the calculation of the initial approximations of the image with an error E in the vicinity minimum value for each cluster number that is absolutely necessary for further error E minimization by the K-means method.

However, in the application to the clustering of a large number of pixels, the following difficulties must be overcome.

First, Ward’s pixel clustering according to [12] takes too long, since the required time depends quadratically on the number of pixels. To solve the problem, it turns out that we need two ways to speed up calculations, namely, by enlarging initial pixel sets and by processing the image in parts. To implement the latter, it is necessary, along with Ward’s pixel clustering, to improve the hierarchies within image parts by splitting/merging sets of pixels to ensure the separability of the joint algorithm.

Secondly, although Ward’s method does not take into account the geometric position of pixels, it, due to multi-iterative calculations, gives different results when changing the scanning order of cluster pairs. Therefore, Ward’s pixel clustering significantly depends on the implementation algorithm, as well as on minor changes in the image, for example, 90-degree rotation.

Thirdly, to ensure the suppression of the dependence of the results of Ward’s pixel clustering, it is required to minimize the error E for a given number $g_{0}$ of clusters, but the K-means method [1,24] is not effective enough for this purpose [25]. The fact is that although it reduces the error E, it does not minimize it, because ends before the possibilities of minimization E are exhausted. This is due to the fact that in the K-means method, in comparison with Ward’s method, the increment of the approximation error is estimated by a coarsened formula. The effect is aggravated when all sets of pixels of hierarchically structured clusters are involved in the minimization process. K-means method, which, unlike the Ward method, is popular in image processing, also has a number of other disadvantages.

Thus, all three clustering methods under consideration, which were developed at one time for manual calculations or with the help of adding machines, need to be modernized as applied to pixel clustering, which we will discuss in the article.

## 3. Definition of Superpixels

*Superpixels* are enlarged pixels, i.e., elementary clusters of pixels that approximate the image and objects in the image with varying accuracy. As the number of superpixels decreases, the approximation accuracy also decreases, but the processing speed increases, which is the main motivation for using superpixels.

The definition of superpixels expresses a formal interpretation of the statement that the optimal approximations of an image consist of superpixels.

Let us consider a series of optimal approximations of a color image in 1,2,…, g colors, where g is given in the range from 1 to G, and G denotes the number of colors in a color image or intensity levels, in the particular case of a halftone image. Then $s\left(g\right)$ *superpixels* in the range from g to g!≡1×2×…×g are calculated as a result of the intersection of g optimal partitions of N image pixels into clusters.

The above definition of superpixels, the example of a standard halftone “Lena” image of 256 × 256 pixels, containing 216 intensities, is illustrated in Figure 2.

Figure 2 for image approximations of 1,2,…, g pixel clusters shows the dependencies of the standard deviation σ on the number of clusters in the image approximations.

The lower gray curve is built for optimal approximations and describes the sequence of the minimum possible values of σ. The upper curve for the current number g of clusters describes the intersection of the sequence of optimal approximations. The sequence of partitions into superpixels resulting from the intersection of optimal approximations is hierarchical. As established experimentally, when σ is converted into the approximation error $E=3N\sigma^{2}$, the lower curve turns to be convex:(1)$$E_{g}\leq \frac{E_{g-1}+E_{g+1}}{2},\;g=2,\;3,\;\ldots ,\;N-1.$$

Unlike the curve for optimal approximations, the curve for superpixels is not convex in the E×g coordinates, and, all the more so, in the σ×g coordinates, as in Figure 2. This limits the accuracy of image approximation by the superpixel hierarchy.

In our model, the sequence of optimal approximations is treated as an *image structural representation*, and it is assumed that it is described by a convex curve for most cluster numbers g. In general, the sequence of optimal image approximations is not hierarchical. So, an image is represented as an *ordered* non-hierarchical structure of approximations, which, with a change in the cluster number, are characterized by a monotonous change in the approximation error E and also by its increment ΔE.

The solid curve in Figure 3 describes the number of superpixels s depending on the number g of optimal approximations that generate the current division of the image into superpixels. In this case, the number of partitions coincides with the number g of pixel clusters in the image approximation.

If the optimal approximations form a hierarchy, then the number of superpixels coincides with the number g of pixel clusters in the current image approximation (dotted curve in Figure 3). However, the actual curve for the number of superpixels deviates markedly from the dotted curve, which makes it possible to quantify how the sequence of optimal image approximations deviates from the hierarchy.

Figure 4 and Figure 5 visually demonstrate image approximations.

A series of eight initial optimal image approximations is shown in Figure 4.

A hierarchical sequence of piecewise constant image approximations by superpixels is shown in Figure 5.

The hierarchy of superpixels in Figure 5 visually looks somewhat better than the hierarchy of optimal approximations in Figure 4, which is explained by the increased number of superpixels compared to the number of clusters in the optimal image approximation.

Numerical characteristics for optimal approximations and approximations of the image by superpixels with the number of pixel clusters from 1 to 20 are given in Table 1.

The first column of Table 1 shows the number g of clusters in the optimal image approximation (Figure 4), which coincides with the current number of considered optimal approximations related to the last two columns. The second column indicates the standard deviation σ of the optimal approximation from the image, which is the minimum possible for a given g. The penultimate column shows the number s of superpixels in the superpixel image approximation (Figure 5), and the last column shows the corresponding standard deviation $\sigma_{s}$.

## 4. Approach to Superpixels

With increasing image resolution, the subject matter of superpixels becomes especially relevant due to the necessity to reduce the computational complexity of image processing by replacing pixel operations with superpixel ones. Therefore, it is widely discussed in the literature, albeit in a non-strict, heuristic formulation of the problem of calculating superpixels as some enlarged pixels, which are usually identified with connected image segments that do not distort the boundaries of visually observed objects [27,28,29,30,31,32,33,34,35,36,37]. There are no generally accepted accurate definitions for superpixels. The calculations use algorithms with unproven or at least non-obvious convergence [27,28]. At best, approximation error reduction methods are used as auxiliary methods, which, however, are often limited to versions of the K-means method. In these cases, the real-life minimization of the approximation error is not performed, and the external criterion for the successful operation of the software is some a priori given samples of the “ground truth”.

Various approaches to the calculation of superpixels are considered in detail in [29]. A stable stereotype is the formation of superpixels in the mode of local pixel enlargement [27,28,29,30,31,32,33,34,35,36,37], although for color images there are also heuristic divisive approaches to determining superpixels [37]. When the pixels are enlarged, the boundaries between the objects are masked by the textural features of the reflecting surfaces of the objects in the scene, which are actually used to select the objects-of-interest in the images of a given subject area. In general, to separate an image into previously unknown objects, it is necessary to suppress texture features, for example, as provided in the definition of superpixels, which is proposed in this paper. In the proposed definition, superpixels are obtained by intersecting the initial optimal partitions of the pixel sets in 1, 2 ,…, g colors, which do not take into account the local textural features of the image that arise with a large number of clusters. It is interesting that according to our definition, the optimal approximations are not the best (Figure 2), and the approach [29] to find the optimal metric solves the problem from the opposite origin. Moreover, when calculating superpixels, excessive minimization of the approximation error can probably have the negative effect of revealing false boundaries between objects.

To accurately calculate superpixels, in addition to their rigorous definition, it is required to specify a method for generating a non-hierarchical sequence of optimal image approximations. In the case of a grayscale image, this is Otsu’s multithreshold method [7], which is a generalization of Otsu’s original method [8] of image binarization. At first glance, the fundamental limitation for the practical calculation of optimal approximations is that the computational complexity of the Otsu multithreshold method grows exponentially with the number of tones in the partition, since the problem is NP-hard. Therefore, using Otsu’s multithreshold method, it turns out to be impossible to obtain optimal partitions in a reasonable time, for example, already in 20 tones. However, the latter circumstance is surmountable due to the fact that, firstly, when calculating the optimal approximations, a rough solution is allowed and, secondly, the solution is considered for a limited set of quite specific image data.

The solvability of the problem of obtaining optimal approximations and the corresponding hierarchy of superpixels is shown in the example of the halftone image “Lena” in Figure 2, Figure 3, Figure 4 and Figure 5 and Table 1. Optimal approximations with a relatively large number of tones are built using methods that, on the one hand, have no more than quadratic computational complexity, and, on the other hand, are applicable not only to grayscale but also to color images.

The practical calculation of superpixels for individual images is useful to find suitable heuristic criteria for pixel merging in high-speed agglomerative pixel clustering methods that are being developed to solve current engineering problems of detection and recognition of objects-of-interest. Experimental verification of the effectiveness of a priori assumptions about the unification of superpixels in terms of the number of pixels, size, shape, and other features, as in [27,28,29,30,31,32,33,34,35,36,37], remains an important research area to refine the concept of superpixels and optimize the calculations, but only to the extent that it does not reduce the efficiency of approximation error minimization.

For the sake of generality of reasoning, we will further agree to consider minimizing the error of approximation of images without using the Otsu method. Then it is possible to solve the problem for color images. The formal definition of superpixels does not change when moving to color images. Since a grayscale image is a special case of a color image, the efficiency of minimizing the approximation error and calculating optimal approximations can be evaluated on grayscale images of three identical color components, and superpixel calculation methods developed for color images can also be tested.

## 5. Model of Hierarchical Approaching of Optimal Approximations

Strictly speaking, *objects* are understood as classes of visually observable objects, i.e., clusters of pixels of certain image approximations, which, with a sufficiently small number of clusters in the approximation, differ in three-dimensional color. Objects and images are assumed to be *structured*, i.e., consisting of objects or images in their turn.

Any set of pixels is considered *structured* if a binary hierarchy of piecewise constant approximations, which is described by a convex sequence of approximation errors is calculated for it.

Obviously, any pixel set, for example, a set of N image pixels or pixel clusters of any image partition, can be considered structured if the required hierarchy of approximations is calculated for each of them. The introduction of structured pixel sets ensures the correct estimation of the increment of the approximation error that accompanies the division of the pixel sets in two. At the same time, the property of convexity of the sequence of approximation errors ensures that the current increment of the approximation error is extremal in comparison with its subsequent increments during the iterative division of a given set of pixels into parts.

For computer detection of objects in an image, the problem of hierarchical approximation of a sequence of optimal approximations is posed and solved. The superpixel hierarchy (Figure 2, Figure 5 and Table 1) can be considered as the required solution without tuning parameters. However, this solution is based on the calculation of the initial series of optimal approximations, for which it is necessary to develop a computational model. In this case, the hierarchical approximation of the sequence of optimal approximations turns out to be ambiguous, and control parameters arise that allow one to adjust to the optimal approximation with a given number of clusters, as well as to control the calculation speed. Whether the developed solution is independent or auxiliary in the calculation of the superpixel hierarchy in Figure 2, Figure 5 and Table 1—experiments will decide in the future.

In the approach under discussion, the image is *structured*, i.e., is approximated by a hierarchy of piecewise constant approximations, which, in contrast to the superpixel hierarchy:is described by a convex sequence of approximation errors, similar to the hierarchy of optimal approximations;is a *binary* hierarchy, where each pixel cluster either coincides with an indivisible superpixel or is divided into two.

Taking into account possible options for approximating the sequence of optimal approximations, hierarchical sequences of approximations are parameterized by the number $g_{0}$ of pixel clusters in the optimal image approximation contained in the target hierarchical sequence.

The model of approximation of a non-hierarchical sequence of optimal approximations by a binary hierarchy of image approximations is illustrated in Figure 6.

Figure 6 describes the hierarchical approximation of the optimal approximations by plotting the dependence of the approximation error E on the number g of pixel clusters in the approximation. In Figure 6, all curves are convex (1) and satisfy the condition:(2)$$\frac{\partial E}{\partial g}\leq 0,$$
where the written value, up to a sign, coincides with the value of the *heterogeneity* parameter: $H=-\frac{\partial E}{\partial g}\geq 0$, which is attributed to the cluster as its integral characteristic and, like the number of pixels in the cluster, does not increase when the cluster i∪ j is divided into its component parts i and j:(3)$$H\left(i\cup \;j\right)\geq H\left(i\right)\;,\;H\left(i\cup \;j\right)\;\geq H\left(j\right).$$

In Figure 6, the lower gray curve describes the sequence of optimal approximations with the number of clusters from 1 to the number N of pixels in the image. The upper black curves describe hierarchical sequences of approximations that approach the sequence of optimal approximations in the ranges of the number of clusters from 1 to s, where s is the number of superpixels in the image.

The thick black curve in Figure 6 describes the target hierarchy of approximations, and the thin black curves describe variants of the hierarchical approximation of optimal approximations. For hierarchical sequences of approximations, the negative increment of $\Delta E_{split}\equiv -H$ as the number of clusters g increases by one describes the division of one of the clusters into two. The parameter $g_{0}$ is chosen in the range from 2 to the maximum value $g_{1}$. Depending on $g_{1}$, the number $s\left(g_{1}\right)$ of superpixels indicates the minimum number of superpixels at which $g_{1}$ of initial optimal approximations without distortion can be obtained by merging s superpixels (Table 1):$$1<g_{0}\leq g_{1}\leq s\left(g_{1}\right).$$

The dashed gray curve in the range of the number g of clusters from $g_{1}+1$ to $s\left(g_{1}\right)$ describes the errors of the hierarchical image approximation by indivisible superpixels when the optimal approximations are calculated with distortions. Distortions for a given g are estimated from below as the difference between the points of the dashed and solid gray curves.

Under the *basis objects* or simply *objects* detected by the computer, we mean the pixel clusters of the optimal image approximation in $g_{0}$ colors, and the number $g_{0}$ of pixel clusters is treated as the number of objects. The number $g_{0}$ is a tuning parameter that can be set by focusing object detection on objects-of-interest. Another independent parameter is either some threshold for the number s of superpixels or the number $g_{1}$ of first optimal approximations that are reproduced without distortion. By means of this parameter, a reasonable compromise is achieved between the processing speed and the detection accuracy of objects-of-interest, which consist of superpixels and can either coincide with the basis objects, or be identified with the union, or with parts of the basis objects.

A feature of the model is that the entire image field is considered to be occupied by objects and describes the scene without “blind” zones. Therefore, to detect objects, total divisions of the set of N pixels into clusters are considered, each of which is treated as a set of pixels of a potential object-of-interest.

The object detection procedure is divided into two stages (Figure 7).

In the first stage, the image is structured by generating an approximation hierarchy (on the left in Figure 7) with the parameter $g_{0}$ (Figure 6). In the second stage, the approximation hierarchy is transformed into a gray representation of the image (on the right in Figure 7), which is designed to visualize the results of the detection of objects-of-interest in a visual form.

The pixel values of the resulting gray image representation are interpreted as automatic designations, labels, which are assigned to color objects detected by the computer in the image depending on the threshold value $H_{threshold}$ of heterogeneity H. In this case, the threshold value $H_{threshold}$ is measured as a percentage of the maximum value and is selected by a software engineer for the convenience of detecting objects-of-interest in the labeled image.

The algorithm for converting the hierarchy of approximations into a gray image representation (*object map*) consists in filling the field of the object map with incremental values of the current markup:all pixels of the feature map are assigned initial zero values, and the current markup value is assumed to be equal to one;clusters i of the hierarchy of image approximations are scanned in order from smaller to larger heterogeneity $H\left(i\right)$;from the number of clusters i with heterogeneity values $H\left(i\right)$ not lower than the established threshold $H_{threshold}$, $H\left(i\right)\geq H_{threshold}$: (a) the next cluster j is selected, marked with zero values on the object map; (b) pixels of cluster j on the object map are assigned the current markup value; (c) the current markup value is incremented by one.

Due to the convexity property (1)–(3), the algorithm is correct, which is not true, for example, if the heterogeneity H is replaced by the average intensity within the pixel cluster. On the other hand, according to the convexity property, instead of the heterogeneity threshold, one can use a threshold based on the number of pixels or the area occupied by cluster pixels.

The main advantage of converting an image into an object map (Figure 7) is that it is controlled by a single threshold value of the $H_{threshold}$ parameter, and not by a pair of values that define a certain range, by which pixel clusters are filtered from a given hierarchy of clusters [38]. At the same time, due to one, rather than the conventional pair of parameters, the online setup of the detection of objects-of-interest is simplified and the total division of the image field into objects is ensured.

The generation of an approximation hierarchy for obtaining a structured image is provided by a system of three modernized methods, namely, Ward’s method, the K-means method, and the split/merge method which are described in the following four sections.

## 6. Recursive Ward’s Method

Some hierarchy of approximations, described by a convex (1) sequence of approximation errors $E_{g=1}$, $E_{g=2}$,..., $E_{g=N}$, is obtained by pixel enlargement by the original Ward method [5,6,12].

In Ward’s method, at first, each pixel constitutes an independent cluster.

Then, at each iteration, a pair of clusters i,j merge with each other, corresponding to the minimum increment of the approximation error $\Delta E_{merge}\left(i,j\right)$:$$i,j\to i\cup j:\;i,j=\underset{i,j=1,2,\ldots g}{\arg\min}\Delta E_{merge}\left(i,j\right),$$
where the number of clusters g decreases from N to 1, and the approximation error increment $\Delta E_{merge}\left(i,j\right)$ is expressed in terms of the number of pixels $n_{i}$, $n_{j}$ in clusters i,j and the three-dimensional average pixel values $I_{i}$, $I_{j}$ within the clusters i,j as:(4)$$\Delta E_{merge}\left(i,\;j\right)=\frac{n_{i}n_{j}}{n_{i}+n_{j}}{\left\|I_{i}-I_{j}\right\|}^{2}\;\geq 0.$$

Figure 8 describes the hierarchy of approximations obtained by Ward‘s method.

In Figure 8, the red solid line shows the dependence of the approximation error E on the number g of clusters (colors) in the Ward’s image approximations, which, in contrast to the black solid curve in Figure 2 is convex and describes a binary hierarchy of image pixel clusters. In Figure 8, the area where the curve is constructed according to Ward’s method is bounded from above by a descending dashed line from the point $g=1,E=E_{1}$ to the point g=N,E=0, and is bounded from below by a solid gray curve, which describes the sequence of optimal image approximations, as in Figure 2.

In this case, for a given number g of pixel clusters, it is guaranteed that successive values of approximation errors $E_{g}$ do not exceed the threshold value written on the right side of the formula:(5)$$E_{g}\leq E_{1}\left(1-\frac{g}{N}\right),$$
where $E_{1}$—approximation error of the image by identical pixels.

Condition (5) expresses the closeness of the pair of curves in Figure 8 as a whole. Despite the fact that both curves in Figure 8 are convex and have a common beginning and ending, they do not coincide, because the curve for optimal approximations describes a sequence of approximations that is not hierarchical. Therefore, the problem of approximating optimal approximations by a hierarchical sequence of approximations seems to be contradictory. The contradiction is eliminated if the required coincidence or at least convergence of the curves is limited to a single point $g_{0}$, as in Figure 6.

The main limitation of the original Ward’s method is that it does not allow one to obtain any hierarchy of image approximations described by a convex sequence of approximation errors. At least, it is not obvious how to obtain the hierarchical approximation sequences provided in the model Figure 6.

This limitation is removed by updating Ward’s method, in which it is applied to image *parts*, i.e., to pixel clusters from a certain image partition. Performing the processing in parts allows not only to approach the optimal approximation sequence according to Figure 6 but also to reduce the computational complexity of the upgraded Ward’s method.

As is known, Ward’s method has computational complexity $N^{2}$ or even $N^{3}$ for head-on programming without storing and converting in the buffer pixel cluster pairs, the merging of which is accompanied by the minimum and closest to the minimum increments of the approximation error $E_{merge}$.

Let the computational complexity C of Ward’s method increase as $N^{\eta }$ when the number N of pixels increases, where η>1. Let us estimate how C changes if Ward’s method is applied *in parts*, i.e., within each pixel (superpixel) cluster, which are first processed as independent images, and then ordered into a binary hierarchy as image elements:$$C=\alpha \cdot g^{\eta }+\beta \cdot g^{1-\eta }\cdot N^{\eta },$$
where α and β are the fixed parameters. Obviously, the written function of g for $g_{0}={\left(\frac{\beta }{\alpha }\cdot \frac{\eta -1}{\eta }\cdot N^{\eta }\right)}^{\frac{1}{2\eta -1}}$ reaches a minimum $C_{\min}$:$$C_{\min}=\alpha^{\frac{\eta -1}{2\eta -1}}\beta^{\frac{\eta }{2\eta -1}}\frac{2\eta -1}{\eta -1}{\left(\frac{\eta -1}{\eta }\right)}^{\frac{\eta }{2\eta -1}}N^{\frac{\eta^{2}}{2\eta -1}},$$
where the formula for $g_{0}$ can be used to automatically calculate the number $g_{0}$ of superpixels from the condition of minimum computational complexity.

Thus, when processing in parts, the computational complexity $C\sim N^{\eta }$ is reduced to $C\sim N^{\frac{\eta^{2}}{2\eta -1}}$.

When recursively repeating the acceleration of calculations in parts, it turns out:

at η=2, C falls off like N2→N43→N1615→N256255→…→Ntt−1....→N, where $t=2^{2i},\;i=1,2,\ldots$;at η=3, C falls off like N3→N95→N8165→N65616305→N43,046,72142,981,185…→N.

Thus, contrary to the prevailing stereotypes, the version of Ward’s pixel clustering by parts with proper programming [39] refers to high-speed image processing methods with almost linear computational complexity.

In addition to the method of structuring the image in parts, the acceleration of calculations is provided by the traditional replacement of pixels by their enlarged sets, in particular, by superpixels.

For the recursive Ward method, in contrast to the original Ward method, it is easy to show the existence of solutions provided in the model in Figure 6. In this case, to approach the image by an approximation hierarchy containing the optimal approximation with $g_{0}$ pixel clusters or superpixels, it is sufficient:to calculate the hierarchy of approximations for each cluster of optimal approximation by the original Ward method, as for an independent image;to rebuild the hierarchy of the image approximations without modifying the calculated pixel clusters, reordering the merging of clusters so that the resulting hierarchy of approximations is described by a convex sequence of approximation errors $E_{g}$, where $g\leq g_{0}$;to complete the hierarchy to the full one, iteratively enlarging $g_{0}$ clusters of the optimal image approximation using Ward’s original method.

Thus, the multivalued hierarchical simulation of optimal approximations in Figure 6 and its parametrization by the number of objects $g_{0}$ are justified. In addition, the automatic calculation of $g_{0}$ parameters from the condition of minimal computational complexity is provided.

The polynomial increase in the computational complexity of the agglomerative algorithm with increasing N means that as N decreases, the computational complexity also falls rapidly. Therefore, in itself, processing by the recursive Ward’s method in parts of the image provides acceleration of calculations. However, if the partitioning of the image into $g_{0}$ clusters and the corresponding approximation are chosen arbitrarily, then the dependence $E_{g}$ on g will turn out to be piecewise convex with a violation of convexity at $g=g_{0}$. This will not happen if, for image partitioning, we choose the image approximation by means of $g_{0}$ structured clusters so that it cannot be improved by the approximation error E by counter operations of splitting one of the pixel clusters in two and merging the pair of other clusters.

The last statement is a necessary and sufficient condition for an approximation to belong to a certain hierarchy of approximations described by a convex sequence of values E. Such an approximation is obtained at the output of processing any image approximation with a fixed number of clusters, by the so-called CI method.

## 7. CI Method for Improving Structured Approximations

*CI (Clustering Improvement)* is the method of improving the quality of image approximations referring to the method of splitting/merging pixel clusters using a reversible cluster merging operation.

Operation (4) of merging clusters i, j into cluster i∪j is considered *reversible* if for each cluster containing more than one pixel, a pair of clusters is stored, by merging which this cluster was obtained in the process of generating the hierarchy: i∪j→i,j. In this case, the division of the cluster i∪j into two is accompanied by a non-positive increment of the approximation error, which coincides with the value of the derivative $\frac{\partial E}{\partial g}$ of E with respect to the cluster number g:(6)$$H\equiv \left|\frac{\partial E}{\partial g}\right|=-\Delta E_{split}\left(i\cup \;j\right)=\Delta E_{merge}\left(i,\;j\right)\;\geq 0,$$
where $\Delta E_{merge}$ is calculated according to (4), and the symbol H (Heterogeneity) denotes the value of the feature of heterogeneity (“complexity” [2], “saliency” [40,41,42,43,44]), which is assigned to each cluster in the considered hierarchy of image pixel clusters and indicates the presence of a noticeable color object in the area of the image occupied by this cluster of pixels.

The CI method is as follows:

At the input, any image approximation with a given number of clusters $g_{0}$ is taken, which are assumed to be structured without loss of generality.

Then:

From $g_{0}$ clusters, such cluster i∪j is selected, the division of which into two is accompanied by the maximum drop maxH in the approximation error E.From $g_{0}$ × $g_{0}$ of cluster pairs, that pair of clusters $i^{\prime }$, $j^{\prime }$ is selected, the merging of which is accompanied by a minimum increment $\min\Delta E_{merge}$ of E approximation error.At $\max H\leq \min\Delta E_{merge}$, the processing ends. If $\max H>\min\Delta E_{merge}$, cluster i∪j is divided into two clusters i and j, and a pair of clusters is merged with a minimum increase in E—either a pair of clusters $i^{\prime }$, $j^{\prime }$, or a new pair of clusters generated by the appearance of clusters i and j. Next, the processing is resumed.

At the output of the CI method, an image approximation by $g_{0}$ clusters is obtained, which is optimized for the approximation error E, and generates a certain hierarchy of image approximations described at the point $g_{0}$ by a convex curve (Figure 9).

Figure 9 graphically describes the approximation error E depending on the cluster numbers g in the image approximations.

The lower gray convex curve still corresponds to optimal approximations. The dotted black curve describes the generation of the initial image approximation of $g_{0}$ clusters in one or another high-speed agglomerative segmentation algorithm [9,10]. The bold arrow shows the decrease in the image approximation error as a result of processing the input approximation with $g_{0}$ clusters by the CI method. The red convex curve describes the hierarchy of approximations, which contains the resulting image approximation by $g_{0}$ clusters, obtained by the CI method. The dashed descending straight line, limits the area of resulting convex $E_{g}$ dependence.

The CI method leaves unchanged any approximation from the hierarchy of image approximations generated by Ward’s method.

The CI method provides the transformation of any algorithm of the iterative merging of pairs of clusters into an image structuring algorithm. At the same time, to obtain an approximation hierarchy described by a convex sequence of approximation error E values, it is sufficient to execute the CI method at each iteration after the merging of a pair of clusters, which can be chosen arbitrarily [21]. True, in this case, it will be necessary to update the hierarchy of nested approximations for image approximations improved by the CI method.

Due to the fact that the CI method guarantees obtaining an image approximation with an approximation error in a limited neighborhood of the minimum achievable values (in Figure 9 below the dashed curve descending from left to right), the effect of improving the approximation as a result of processing by the CI method is the more pronounced than the initial approximation is rougher [22].

Since the CI method minimizes the increment of the approximation error for cluster pairs, its computational complexity with an increase in the number $g_{0}$ of clusters increases quadratically. However, as $g_{0}$ increases, the approximation error itself drops sharply, which makes it possible for large $g_{0}$ to use the CI method in the segmental version, i.e., in the form of the SI method (Segmentation Improvement) [45] with a linear increase in computational complexity.

## 8. K-Meanless Method for Improving Structured Approximations

To obtain an optimal or at least suboptimal approximation according to the model in Figure 6, the processing of structured image approximation with $g_{0}$ clusters in Figure 9 continues with the so-called K-meanless method [14], which is illustrated in Figure 10.

In Figure 10, the black vertical arrow shows the minimization of the approximation error E of the image approximation following the minimization E performed by the CI method (Figure 9). As a result of processing, an optimal or at least optimal-like image approximation is obtained, which generates a target hierarchy of approximations described by the red convex curve. The target approximation hierarchy approaches the sequence of optimal approximations and contains the optimal image approximation with $g_{0}$ pixel clusters. When the parameter $g_{0}$ changes, the minimization of E by K-meanless method ensures the calculation of other optimal or optimal-like approximations.

Like the K-means method, K-meanless or “K-means without means” method [14] reduces the approximation error E by reclassifying pixels from one cluster to another cluster. At the same time, the fundamental difference is the reclassification criterion in the K-meanless method, which is reduced to the condition of a negative $\Delta E_{correct}$ increment that accompanies the reclassification of pixels from cluster to cluster:(7)$$\min\Delta E_{correct}<0,$$
where the minimum is computed over a set of clusters for increments of the approximation error that accompanies the reclassification of each part from a given cluster to every other cluster.

The $\Delta E_{correct}$ increment of the approximation error during the reclassification (transfer) of k pixels from cluster 1 to cluster 2 is expressed by the formula:(8)$$\Delta E_{correct}=k\left(\frac{n_{2}}{n_{2}+k}{\left\|I_{2}-I_{k}\right\|}^{2}-\frac{n_{1}}{n_{1}-k}{\left\|I_{1}-I_{k}\right\|}^{2}\right),$$
where $n_{1}$, $n_{2}$, $k<n_{1}$ and $I_{1}$, $I_{2}$ are the values of the number of pixels and three-dimensional average intensities in the donor and acceptor clusters, and $I_{k}$ is the average value of k pixels, which are excluded from the number of pixels of the first cluster and assigned to the second cluster.

The formula for $\Delta E_{correct}$ follows from (4) and transforms into the corresponding coarsened analytical expression for the K-means method under the assumption that the number k of pixels is negligibly small compared to the values $n_{1}$, $n_{2}$ of the pixel numbers in the donor and acceptor clusters. However, the use of the approximate formula of the K-means method instead of the exact Formula (8) for $\Delta E_{correct}$ practically does not complicate calculations on a modern computer. Therefore, the use of the basic simplified formula in numerous versions of the K-means method [15,16,17,18,19] is not sufficiently justified.

For the efficient E reduction by the K-meanless method, the order of transformation of triples of clusters that ensure the reduction of E is important. When processing hierarchically structured images to effectively minimize the approximation error E according to criterion (7), the reclassification of pixel sets is performed from large to small, and individual pixel reclassification, which is often only conducted in K-means, is performed last and provides the final minimum optimization approximation. In this case, the assumption that k is negligibly small compared to $n_{1}$ and $n_{2}$ turns out to be incorrect. In order to avoid calculating false minima E, in most cases it makes sense to replace the K-means method with a stronger K-meanless method, in which the reclassification of pixel sets is performed either by comparing the values of the objective functional E, as in [14], or by the equivalent Formula (8) for $\Delta E_{correct}$, as in [13].

## 9. Discussion of the System of Methods for *E* Minimization

Three methods, namely, the recursive Ward’s method in parts, the CI method, and the K-meanless method, applicable to both grayscale and color images, are the basis of the hierarchical approximation model in Figure 6 of optimal image approximations with minimization of the approximation error E. It is essential that the listed methods are analytically derived from the requirement of minimizing the additive functional E and constitute a system of methods since the use of each individual method is not effective enough to minimize E.

Ward’s recursive method by parts of the image, when approximating the optimal approximations, minimizes the approximation error E for the binary hierarchy of pixel clusters as a whole due to the fact that the target hierarchy of pixel clusters, like the sequence of optimal image approximations, is described by a convex sequence of E values.

The CI method and K-meanless method provide minimization of E for image approximation with a fixed number $g_{0}$ of pixel clusters. The CI method is used to optimize the partition into $g_{0}$ clusters during the execution of Ward’s method, as well as to reduce E before the K-meanless method executing. The k-meanless method minimizes the error when changing the parameter $g_{0}$. A number of initial hierarchical image approximations in a limited $g_{0}$ number of colors determines the partitioning of the image and computer-detected color objects into superpixels.

The optimal (suboptimal) approximation of an image with $g_{0}$ hierarchically structured clusters, described by a convex sequence of approximation error values E, is determined by the fact that it cannot be improved in E by:The CI method, i.e., by means of counter operations of splitting one of the clusters in two with the subsequent merging of a pair of other clusters;The K-meanless method, i.e., reclassification operation, partially reassigns the pixels from one to another cluster.

The given optimality conditions are constructive since their violation indicates a way to compensate for this violation by minimizing E. However, it should be taken into account that in the process of minimizing E, the condition of structuredness of pixel clusters, generally speaking, is violated. Specifically, the minimization of E is performed using two operations-splitting a pixel cluster in two and merging a pair of pixel clusters into one cluster. If the given cluster is structured, then, when split in two, it is transformed into a pair of structured clusters. However, a pair of structured clusters may turn out to be unstructured upon merging due to the violation of the convexity property of the resulting sequence of approximation errors. When programming, the easiest way is to get around the formulated problem. To do this, it is sufficient to supplement the operation of merging pixel clusters by detecting and suppressing the violation of the convexity property of the errors E sequence by updating the broken hierarchical structure of the cluster. If we do not resort to the composite operation of merging pixel clusters, then the desired result is achieved by cyclically repeating the process of minimizing E until it stabilizes.

In addition to preserving the hierarchical structure within the clusters, in the course of calculations, it is necessary to maintain the smoothness of the $E\left(g\right)$ dependence at the $g_{0}$ value, periodically repeating the processing by the CI method of the current image approximation in $g_{0}$ colors. As in the case of the merge operation, the correction of approximations by the CI method can be performed either online or cyclically repeated until the possibilities of minimizing E are exhausted.

## 10. Dynamic Table of Ordered Image Approximations

This section describes the implementation of the discussed model as some computer program that helps the user or programmer control the detection of *hierarchically structured* objects containing nested objects.

Figure 11 shows a graphical representation of an image as a superposition of N hierarchies of pixel clusters, described by convex sequences of approximation errors $E_{g}$ depending on the color numbers g.

Calculation of optimal approximations in Figure 11 is necessary for the full implementation of the image model in Figure 6, providing the generation of the superpixel hierarchy.

NOTE. Sequences of optimal approximations and object hierarchies Figure 11 can be obtained using only Ward’s original method, which is applied to enlarged pixels, if the pixel enlargement, for example, by segmentation, does not affect the perception of the image. In this case, first, Ward’s method is applied, say, 100 times with a different number of enlarged segments. Then, from the resulting hierarchies of pixel clusters, a subsequence is selected that provides the minimal standard deviations in a given range of the color numbers, for example, from 1 to 15.

In the case of using enlarged image pixels, in particular, superpixels from Figure 6, one can also use the scheme of Figure 11, in which the number of pixels N is simply replaced by a smaller number of enlarged pixels if the original image is replaced with its real-valued tri-component color representation obtained by averaging the pixels within their enlarged sets.

Figure 11 describes the set of $N^{2}$ image approximations that should be available to the user. In this case, the program must provide access to one or several hierarchies of image approximations in some range of colors. To ensure this, it is convenient to arrange the approximations of Figure 11 in the form of a Dynamic Table of approximations, which are fragmentarily calculated online.

Let us explain the concept of a Dynamic Table using an example of a composite image taken from [46] (Figure 12).

The image in Figure 12 contains 10 nested ones, including an image of a surfer (second from the left), whose body detection in the bottom-up strategy proves to be problematic.

The Dynamic Table of approximations for the above image is obtained in the form of Figure 13.

The 4 × 4 fragment of Dynamic Table of (1774 × 272)^2^ = 232,833,270,784 image approximations.

In the Dynamic Table, hierarchical sequences of approximations are arranged in columns. The rows show image approximations in the same number of colors. On the main diagonal are the optimal or optimal-like image approximations, with minimum standard deviations within the rows. If the minimums for different rows fall into one column, then this column is repeated in the Dynamic Table (to reduce the width of the table, repeated columns are excluded in Figure 13).

Image approximations in the Dynamic Table are ordered along the columns and along the main diagonal. This means that the sequences of approximation errors $E_{g}=3N\sigma^{2}$ in these directions are convex. 

Table 2 lists the standard deviations of the image approximations. The squares of the tabulated σ values form convex sequences along the columns and the main diagonal.

NOTE. The property of convexity of the sequence of $E_{g}$ values in columns is always provided by the construction algorithm. A significant corruption of the convexity along the main diagonal indicates either insufficient minimization or a violation of the basic model assumption about the specifics of the image data. The study of counterexamples of images and other data characterized by a pronounced non-convex dependence of approximation errors $E_{g=g_{0}}$ along the main diagonal is of independent interest.

Diagonal optimal or optimal-like image approximations are resistant to algorithms for their calculation since optimal approximations are determined only by the minimum values of approximation errors $E_{g}$, and not by a specific calculation algorithm. Therefore, it is the optimal image approximations that are convenient to use to select the desired cluster hierarchy for best detecting structured objects-of-interest.

So, the procedure for detecting structured objects-of-interest may be that the user first analyzes the diagonal optimal approximations, which consist of basic objects. In overdiagonal image approximations of the image, objects are represented by unions, and in underdiagonal approximations, by parts of basis objects. The user chooses the parameter $g_{0}$ and the hierarchy of clusters from among the available ones in such a way that in a certain range of color numbers g it is better than others to model the hierarchy of objects-of-interest in accordance with his perception.

Through a good choice of approximation hierarchy, one can focus the system on the target objects-of-interest and, for example, overcome the loss of objects in [46] (Figure 14).

Figure 14 in the bottom line shows the 14th approximation of the test image (Figure 12) from the hierarchy in Figure 13 containing the optimal-like approximation in three colors ($g=g_{0}=3,\sigma =30.5178$). In this approximation, the surfer’s body (encircled by dotted line) stands out mainly in three colors. In other hierarchies in Figure 13, background spots on the surfer’s body are revealed.

After selecting the hierarchy, object detection continues as in Figure 7. Alternatively, the image is replaced by a representation in a reduced number of colors, as in Figure 14, and processing is continued by suitable known methods.

The hierarchical representation of the image provides a number of additional features, such as: ranges of hierarchy levels, i.e., values of g within which the cluster or segment do not change; sequences of approximation errors E and their increments ΔE calculated for clusters and segments at different hierarchical levels g, etc. The calculation of additional features for image pixels as elements of clusters and segments contributes to automating the detection of objects and object hierarchies.

It seems promising to implement the proposed variants of the three classical methods of cluster analysis in the widely used MatLab software package.

## 11. Conclusions

Thus, in this paper, we have considered the informal component of the model of a digital image, its elements (superpixels), and objects in the image, developed for automatic detection and recognition of objects. It turns out that in order to unify and, most importantly, simplify the detection and recognition of objects, it is useful to pre-order image data at the primary stage of image processing, which is provided by a system of modernized cluster analysis methods.

It should be noted that in addition to the described conceptual model, its computational version has been developed. In the computational model, high-speed calculations are performed in terms of the so-called algebraic multilayer network (AMN). Characteristically, for hierarchical data ordering, AMN uses Sleator–Tarjan dynamic trees instead of conventional trees. This speeds up calculations and saves memory. The main advantage of Sleator–Tarjan dynamic trees is the most concise description of the splitting/merging pixel clusters by breaking/setting the arcs connecting the image pixel coordinates. The latter helps to simplify the software implementation of reversible calculations; however, the network AMN data structure only optimizes computations but does not change the conceptual model in any way. Therefore, we leave its detailed description for subsequent papers.

We are working on the implementation of a model for solving the engineering problems [47,48]. However, this is not the main thing, as modern image processing contains a variety of different engineering solutions. The main thing is to understand and, therefore, surpass the unified natural visual perception. Perhaps we are on the right track.

## Figures and Tables

**Figure 1 jimaging-08-00274-f001:**
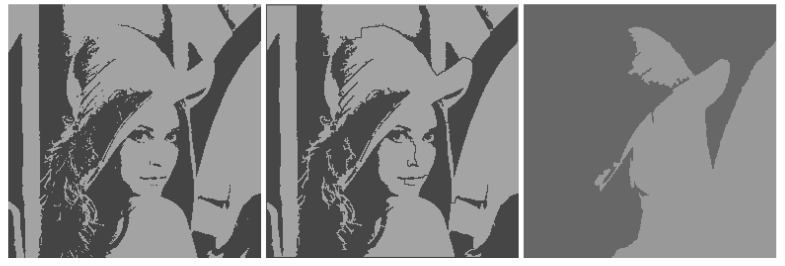
Optimal (on the left), really optimized two-segment approximation (central), and iteratively segmented (on the right) image approximation with the standard deviations *σ* =30.64564, *σ* = 31.60341 and *σ* = 50.33156.

**Figure 2 jimaging-08-00274-f002:**
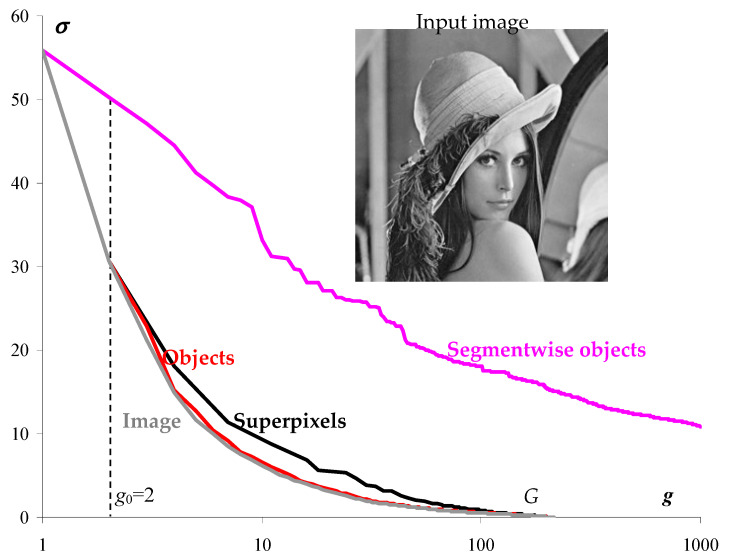
Structural representations of image, objects, and superpixels. The plots illustrate the standard deviation *σ* of approximations from the image depending on cluster number *g*. The upper purple curve describes the hierarchical segmentation of the image. The black solid curve describes the superpixel hierarchy. The lower gray curve describes the sequence of optimal image approximations, and the red curve describes the object binary hierarchy of approximations obtained by the hierarchical Otsu method [9,10,26].

**Figure 3 jimaging-08-00274-f003:**
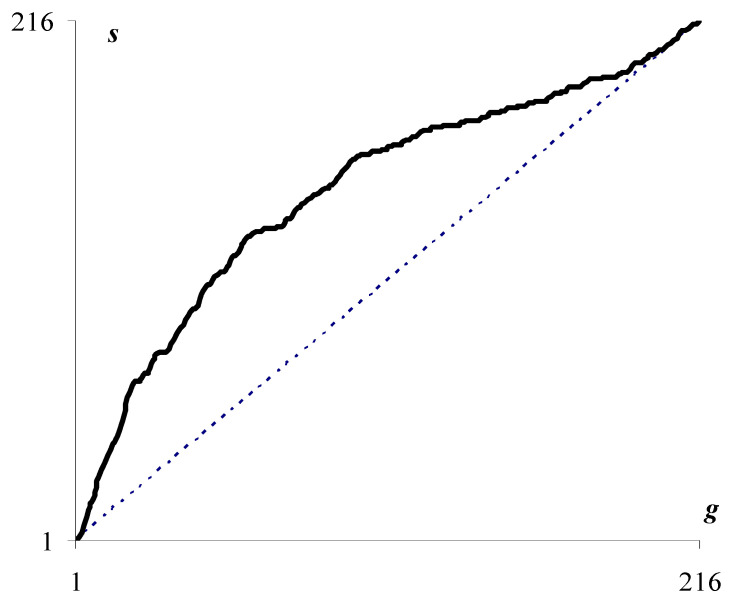
The number s of superpixels as a function of the number g of optimal image partitions involved in dividing N pixels into superpixels due to the accumulation of boundaries between clusters. The dotted curve shows the case of hierarchical optimal approximations.

**Figure 4 jimaging-08-00274-f004:**
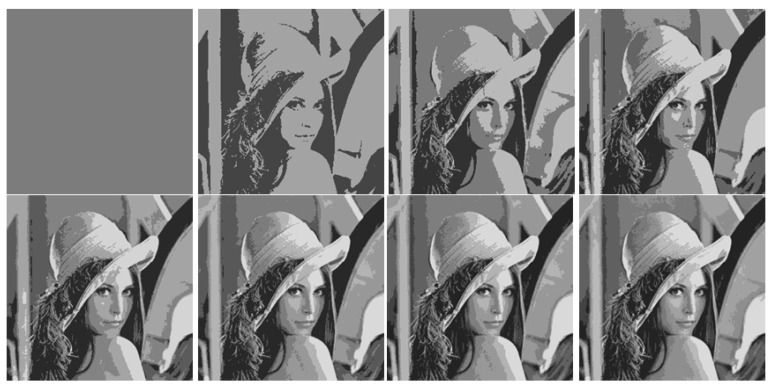
Optimal approximations of a standard image in 1–8 tones, ordered from left to right and top to bottom.

**Figure 5 jimaging-08-00274-f005:**
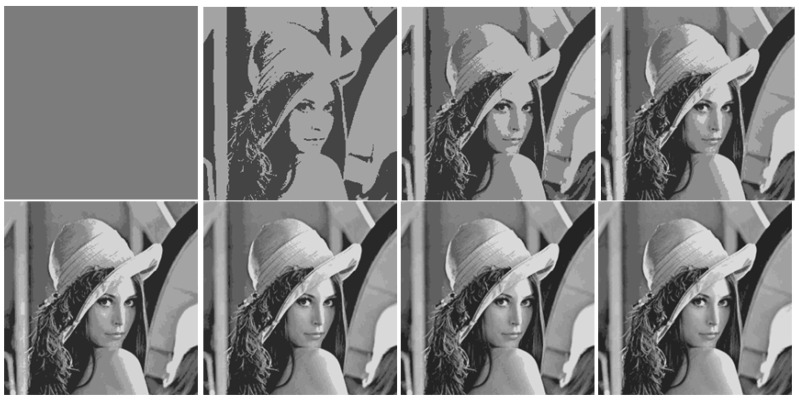
Hierarchy of superpixels generated by eight optimal approximations, ordered from left to right and top to bottom.

**Figure 6 jimaging-08-00274-f006:**
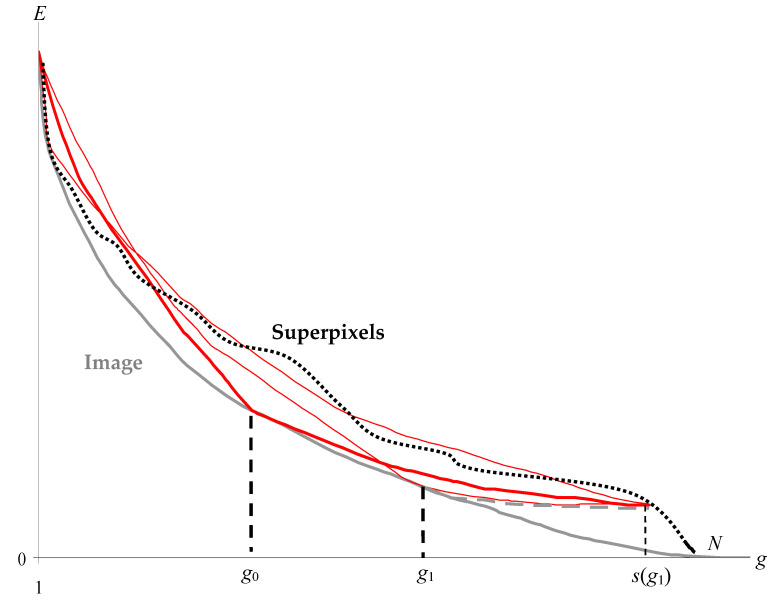
Model of hierarchical approaching of optimal approximations. The limiting lower gray curve, assumed to be predominantly convex, describes the optimal image approximations. The bold red convex curve tangent to the gray curve at *g* = *g*_0_ describes the target hierarchy of image approximations. Another thin red convex curve tangent to the gray curve describes the target hierarchy of approximations at the maximum value *g*_0_ = *g*_1_. The rest upper thin red convex curve describes the hierarchy of approximations produced by Ward’s original method. The dotted black curve describes the hierarchy of superpixel approximations or enlarged pixels intended to initialize Ward’s pixel clustering.

**Figure 7 jimaging-08-00274-f007:**
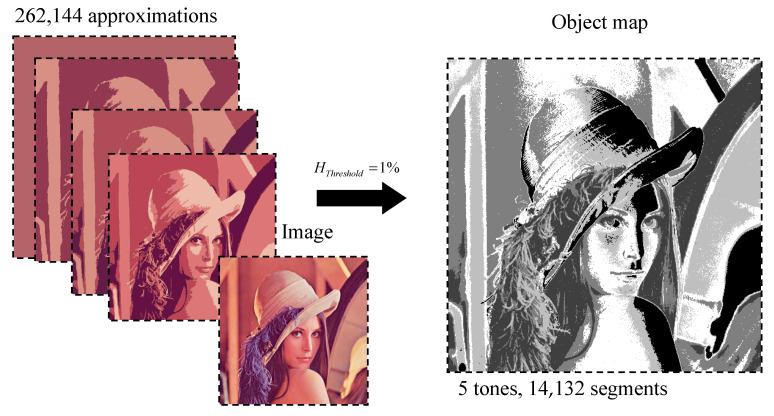
Two-stage detection of objects in a color image.

**Figure 8 jimaging-08-00274-f008:**
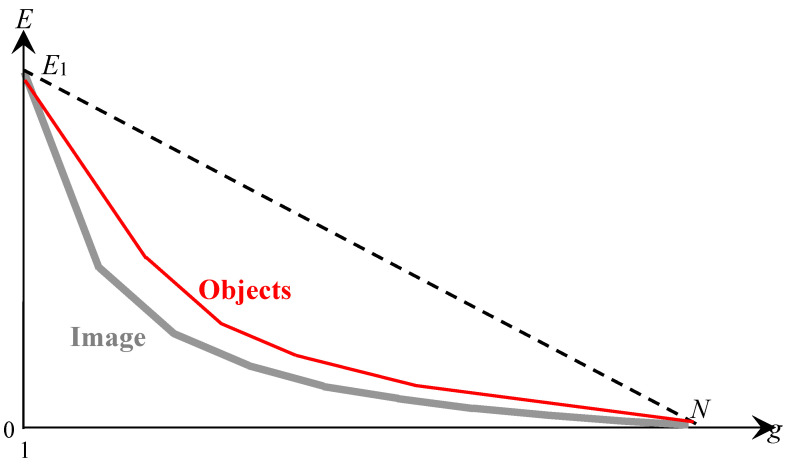
Hierarchical approaching of optimal approximations according to original Ward’s method. The limiting lower gray curve, assumed to be predominantly convex, describes the minimal approximation errors $E_{g}$ of the optimal image approximations depending on the number g of colors. The red solid convex curve describes approaching the image by the hierarchy of approximations for Ward’s pixel clustering. The dashed line shows the upper limit for both error sequences.

**Figure 9 jimaging-08-00274-f009:**
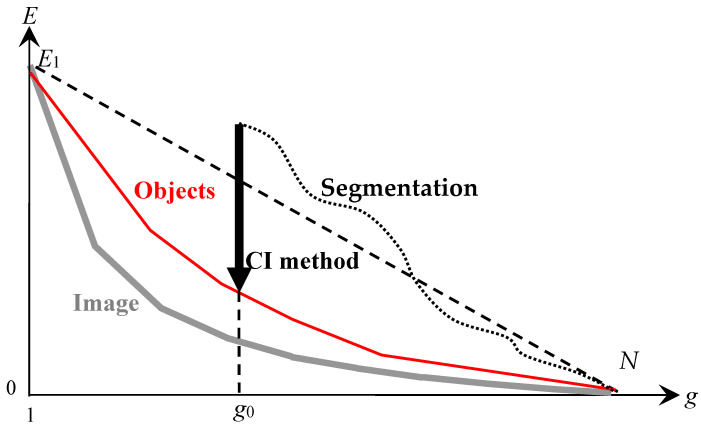
The CI method for minimizing the approximation error with the same number of clusters in the image approximation. The limiting lower gray curve, which is assumed to be predominantly convex, describes the optimal image approximations. The upper dotted curve describes the generation of some approximation of the image, specifically, by hierarchical segmentation. The intermediate convex red curve describes the resulting hierarchy of approximations obtained by the CI method in combination with Ward’s method.

**Figure 10 jimaging-08-00274-f010:**
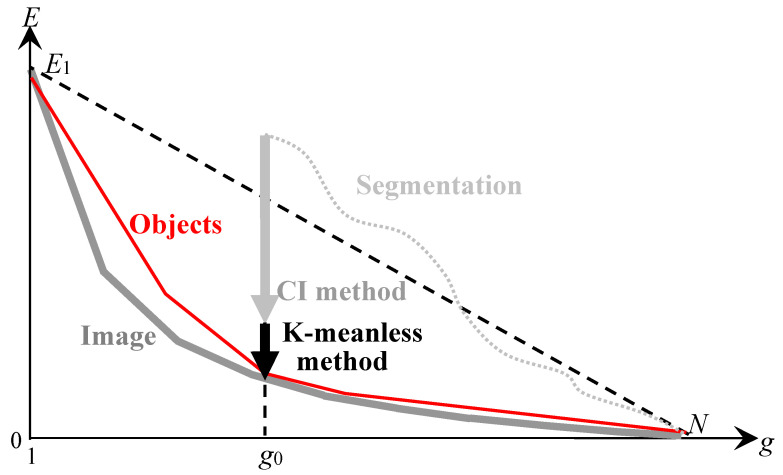
K-meanless method of minimizing the approximation error for a constant number of clusters in image approximation. The limiting lower gray curve, treated as predominantly convex, describes the optimal image approximations. The red convex curve tangent to the gray curve describes the hierarchy of approximations obtained using the K-meanless method.

**Figure 11 jimaging-08-00274-f011:**
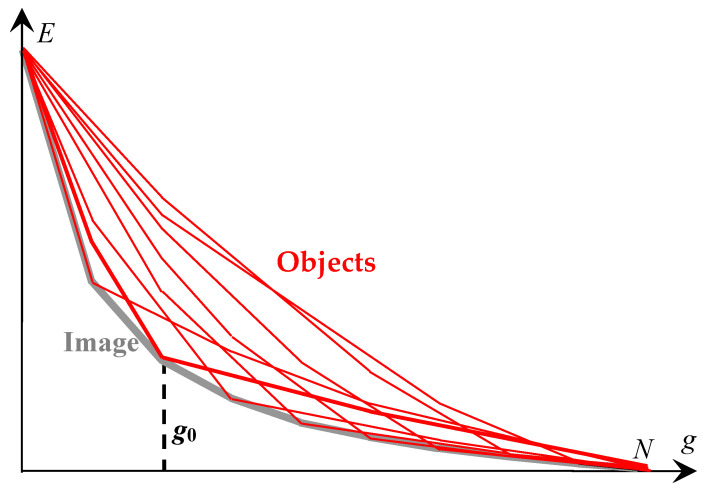
Parameterized approaching of an image by a hierarchical sequence of approximations in *g*_0_ = 1, 2,..., *N* colors. The lower gray convex curve describes *E_g_* sequence of optimal image approximations. The remaining red convex curves describe the hierarchies *E_g_* sequences of image approximations each containing at least one optimal approximation, in *g*_0_ = 1, 2,..., *N* colors.

**Figure 12 jimaging-08-00274-f012:**
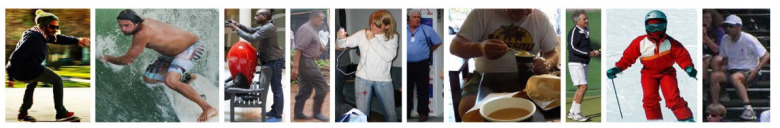
Test image.

**Figure 13 jimaging-08-00274-f013:**
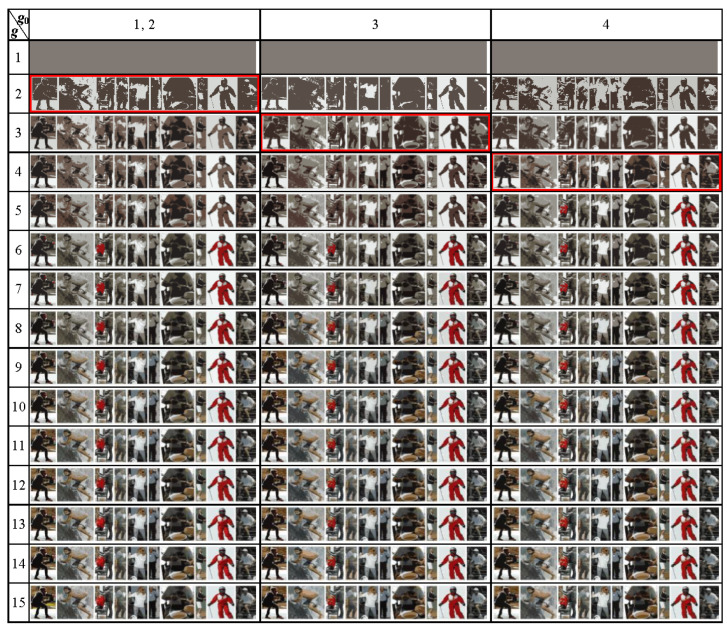
Dynamic Table of *N × N* image approximations, ordered along the columns and the main diagonal. Diagonal elements are highlighted in red. Dynamic Table demonstrates the hierarchies of image approximations arranged in columns.

**Figure 14 jimaging-08-00274-f014:**
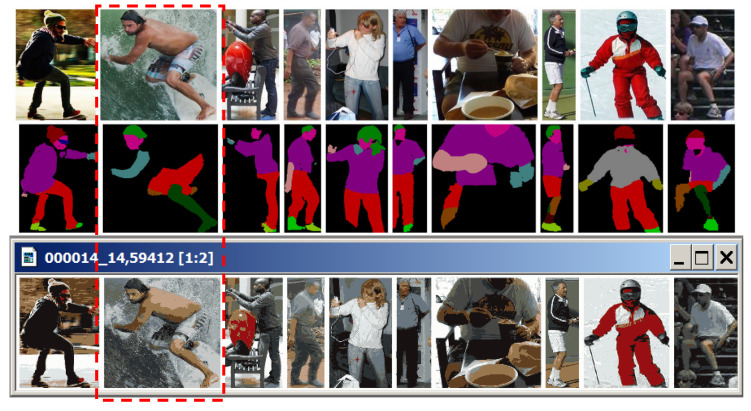
Customizable selection of desired object hierarchy. The input composite image from [46] is displayed in the top line. The middle line shows the result of object detection according to [46], where the surfer’s body is poorly detected. The bottom line shows the proposed method results using *g*_0_ = 3, *g* = 14.

**Table 1 jimaging-08-00274-t001:** Characteristics of optimal and superpixel approximations ^1^.

g	σ	s	σ_s_
1	55.8832	1	55.8832
2	30.6456	2	30.6456
3	21.2174	4	18.1409
4	14.9645	7	11.3062
5	11.6976	11	8.71359
6	10.0398	16	6.85761
7	8.46072	18	5.67555
8	7.51121	24	5.27755
9	6.81359	27	4.62883
10	6.14397	30	3.84431
11	5.57864	33	3.71779
12	5.11403	36	3.18455
13	4.75689	39	3.09882
14	4.42306	41	2.91261
15	4.17825	43	2.61455
16	3.92460	46	2.27893
17	3.70326	50	2.10761
18	3.50441	55	1.95239
19	3.32383	60	1.68547
20	3.15658	63	1.64135

^1^ A complete table of standard deviations of optimal approximations for the Lena image is published in [26].

**Table 2 jimaging-08-00274-t002:** The 15 × 15 fragment of Dynamic Table of (1774 × 272)^2^ = 232,833,270,784 *σ* values denoting image approximations ^1^.

	g_0_	1, 2	3	4	5, 12	6	7, 9	8	10, 11	13, 14, 15
g										
1		**82.7624**	82.7624	82.76244	82.7624	82.7624	82.7624	82.76244	82.7624	82.76244
2		**41.9853**	45.6371	44.61491	48.8133	47.1703	47.1703	47.5201	46.2158	49.70901
3		34.1905	**30.5178**	32.17524	31.9416	32.6948	31.0402	31.40646	30.7935	32.2961
4		27.4511	27.4203	**26.25674**	27.99	28.3736	26.4501	27.7935	27.1017	27.13618
5		24.1529	24.3494	23.98987	**23.4904**	23.5732	23.8579	24.41134	24.5259	23.81052
6		21.5788	21.9211	22.23782	21.1801	**20.922**	21.2423	21.8049	21.9532	21.05243
7		20.2058	19.8155	20.65018	19.9869	19.769	**19.6739**	19.9542	20.1168	20.00413
8		18.744	18.5328	19.08621	18.9059	18.7525	18.443	**18.39776**	18.644	18.93809
9		17.5868	17.6728	17.99386	17.7844	17.8988	**17.297**	17.46976	17.3337	17.94591
10		16.6735	16.8386	17.05136	16.726	17.0394	16.7309	16.80458	**16.514**	16.92667
11		15.8308	16.1807	16.34836	15.8691	16.3098	16.1497	16.1853	**15.788**	15.95407
12		15.3293	15.553	15.86705	**15.2115**	15.7082	15.6135	15.59814	15.2757	15.2257
13		14.834	15.0178	15.37627	14.7058	15.0876	15.1058	15.10836	14.7567	**14.67922**
14		14.3752	14.5941	14.91152	14.2989	14.5841	14.603	14.62753	14.2415	**14.20433**
15		13.9324	14.1667	14.49436	13.9219	14.0773	14.0968	14.15316	13.7995	**13.79826**

^1^ The diagonal elements are highlighted in bold and are listed in the first table row.

## Data Availability

Used in the paper and other useful program codes are available at: https://disk.yandex.ru/client/disk/ExperimentalFindingsKhar (accessed on 28 September 2022). Requests for advice on the operation of programs are welcome.

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
