# Peer review of "A Model of Pixel and Superpixel Clustering for Object Detection"

_2313-433X, 2022, doi:10.3390/jimaging8100274_

Round 1

Reviewer 1 Report

1. Write more detail of super-pixel in introduction section

2. Make a simple flow chart of your proposed method to better understand the methodology.

3. Related work is missing, please add related work (https://www.sciencedirect.com/science/article/pii/S0957417420304784)

4.  add more results and compare your method with other state of the art superpixel methods. better is to add comparison table. 

5. Is your method computationally efficient ? how?

6. what is the limitation of your method ?

Reviewer 2 Report

1. The title is a bit too general, how exactly it this manuscript performing superpixel clustering? I recommend to a few details regarding the main concepts proposed.

2. In the abstract section, after the definition of superpixels, it’s not clear what is the connection between the following sentences and the rest of the abstract section. The last sentence makes a bit of light but is literally at the end of the section.

3. End of abstract section. How is this method comparing with state-of-the-art methods? Please add a sentence regrading this issue.

4. Please highlight and claim the novel contribution of the manuscript in the abstract section. Which ones of these concepts (presented in the abstract) are novel? 

5. I would only recommend to modify “orthodox requirement” as it might not be well understood by the reader.

6. Please add citations the methods listed in lines 46-47.

7. Before the manuscript outline paragraph, please highlight what are the proposed novelties of this manuscripts.

8. Figure 1, please add a legend to the figures. What are black and grey curves representing? Please add some space between the figure and it’s caption.

9. Figure 2 is a bit too simple. Why not add a few more labels to the two axes? 

10. The information in lines 109-111 should be added also in Figure’s 3 caption.

11. The title of section 3 is a bit too general. I recommend to include keywords such as “review”, “state-of-the-art”, etc.

12. It’s not clear from where the description of the proposed method starts. Is it Section 4?

13. Figure 6. Please add a legend to the figure. What are each curve representing? There is also a small typo at the bottom of the figure.

14. End of line 252: typo.

15. Line 336, please check again the correctness of the equation. 

16. Lines 337 and 383: there is a flying “g”.

17. Figure 8: another quite simple image, no legend, not that much information regarding what it represents, without reading the section’s text.

General comments:

I. The authors must clarify what exactly are the proposed novel contributions of this manuscript.

II. From the manuscript structure is not clear what exactly is the proposed method.

III. The figures are a bit too simple and do not highlight enough the ideas behind. Several extra details must be added.

IV. The manuscript must provide and discuss a comparison with state-of-the-art methods. 

V. There a still some small types that must be corrected.

I recommend the authors to clarify these points and modify the manuscript accordingly.

Round 2

Reviewer 1 Report

Thanks for improving my comments. I don't have more comments 

Reviewer 2 Report

  1. Answer to comment 2. The abstract is a bit too long, it is recommended to contain around 200 words. There are also quite many details. It’s hard to follow what exactly the manuscript is proposing.
  2. Answer to comment 3. Please add an explanation at the end of the abstract and claim such novelty. Add a sentence regarding the state-of-the-art methods.
  3. Answer to comment 4. The list of novelties is quite long. At least the last two can be dropped. It’s not clear to me what the authors want to express by “We offer analytically sound modernized versions of three classical methods […]”.
  4. For controlled detection of structured objects, it is proposed to solve an approximation-optimization problem of ambiguous simulating of a non-hierarchical sequence of optimal image approximations by image approximation hierarchy that contains one of them in […]” very hard to read, the sentence is too complex and hard follow. Similar comments to the other bullets.
  5. Answer to comment 8. Thank you for adding the explanation in the figure’s caption, however, please add a legend to each plot.
  6. Answer to comment 12. I’m asking the question because the manuscript itself does not provide the answer, not just for me to know the answer. Please modify the manuscript in such a way that such information can be easily found.
  7. In my opinion, the reason why we one can’t detect where the description of the proposed method starts is that there are quite a lot of details. The information must be structured in such a way that it is easy to understand and easy to follow. The chain of ideas is not clear.
  8. Answer to comment 13. The legend is still missing, only the figure’s caption was updated with information. It’s hard to distinguish between the curves, please used color.
  9. Similar comments as above for figure 11.
  10. Line 107: “Section 2 argues the advantage of […]” doesn’t sound well in English, it is probably a direct translation.
  11. Please improve the structure of the manuscript. Please rephrase maybe lines 107-120. Provide a better connection between the sections. How are all these sections describing the proposed method? Why the information in Section X is important? For which part of the method?
  12. Table 1 caption: It’s missing information regarding how this data was obtained.
  13. Figure 13. Please add in the figure more information regarding each line of images.
    1. Focusing the program on detecting the desired object hierarchy.”, please rephrase.
    2. From above is the image from [46].”, please rephrase, e.g.: “(Top line) Input images from [46].”
    3. In the middle is the result of detecting the objects where the surfer's body was not successfully detected.”, please rephrase, e.g.: “(Middle line): Object detection results using method X …”.
    4. Below is our image approximation in 14 colors for the parameters g0=3, g=14.”, please rephrase, e.g.: “(Bottom line): Proposed method results using g0=3, g=14
  14. Please rephrase the last paragraph in Section 10. “modernized versions”? “public MatLab application package” – MATLAB is not a publicly available software. “[…] it is possible that users and programmers themselves will figure out how to effectively use the Dynamic Table.
  15. Please rephrase the conclusions section. It is not recommended to start a paragraph with “So”. “[…] we have considered the informal component of the model of digital image, its elements (superpixels) and objects in the image, developed for automatic detection and recognition of objects.” Please improve the structure of the sentence. Highlight the idea that you want to present. It’s not recommended to add citations in a conclusions section. The last sentences seem to present feature work rather than conclusions for this manuscript.
  16. More is not always better. The quality of the presentation is very important so that anyone can understand the proposed ideas so that they can use them (and cite) in another research. Please improve the quality of the manuscript. The presentation must follow a chain of ideas.

Round 3

Reviewer 2 Report

The quality of the manuscript was improved, and many details were added. There are a lot of other quite small details which I wish they were still corrected. However, I consider that the current version of the manuscript differs quite significantly from the initial submission, and the authors invested quite a lot of effort in modifying it.

 Therefore, I recommend the editors to accept this manuscript for publication.